# Lineage Plasticity in Cancer: The Tale of a Skin-Walker

**DOI:** 10.3390/cancers13143602

**Published:** 2021-07-18

**Authors:** Archana P. Thankamony, Ayalur Raghu Subbalakshmi, Mohit Kumar Jolly, Radhika Nair

**Affiliations:** 1Cancer Research Program, Rajiv Gandhi Centre for Biotechnology, Kerala 695014, India; archanapt@rgcb.res.in; 2Manipal Academy of Higher Education (MAHE), Manipal 576104, India; 3Centre for BioSystems Science and Engineering, Indian Institute of Science, Bangalore 560012, India; subbalakshmi@iisc.ac.in

**Keywords:** lineage plasticity, tumor progression, metastasis, therapy resistance, epithelial-mesenchymal plasticity

## Abstract

**Simple Summary:**

Lineage plasticity is the ability of cells to transform from one cell type to another and is an important part for tissue repair and for maintenance of homeostasis. Unfortunately, the very same processes can be corrupted in cancer when the molecular checkpoints controlling the process are compromised; as a result of which treatment resistance and disease recurrence can emerge. It can be triggered by treatment received and has been seen across solid and liquid tumors. This review discusses the factors that control different manifestations of lineage plasticity in various cancer types and more importantly, discusses ideas to potentially revert this phenomenon.

**Abstract:**

Lineage plasticity, the switching of cells from one lineage to another, has been recognized as a cardinal property essential for embryonic development, tissue repair and homeostasis. However, such a highly regulated process goes awry when cancer cells exploit this inherent ability to their advantage, resulting in tumorigenesis, relapse, metastasis and therapy resistance. In this review, we summarize our current understanding on the role of lineage plasticity in tumor progression and therapeutic resistance in multiple cancers. Lineage plasticity can be triggered by treatment itself and is reported across various solid as well as liquid tumors. Here, we focus on the importance of lineage switching in tumor progression and therapeutic resistance of solid tumors such as the prostate, lung, hepatocellular and colorectal carcinoma and the myeloid and lymphoid lineage switch observed in leukemias. Besides this, we also discuss the role of epithelial-mesenchymal transition (EMT) in facilitating the lineage switch in biphasic cancers such as aggressive carcinosarcomas. We also discuss the mechanisms involved, current therapeutic approaches and challenges that lie ahead in taming the scourge of lineage plasticity in cancer.

## 1. Introduction

Maintenance of cellular identity is a defining feature of metazoan evolution [1] (Box 1). During embryonic development, diverse cell types are generated from stem/progenitor cells through differentiation. According to the central dogma of embryology, the process of cell fate determination and differentiation is irreversible and unidirectional [2]. After each subsequent step of cell differentiation, the cells become increasingly lineage restricted, resulting in a final mature cell type. Conrad Hall Waddington envisaged this idea metaphorically as a ball rolling down the hill which can take multiple pathways to distinct fates [3]. Once a fate has been chosen, it cannot be reverted. However, this traditional, unidirectional view of cellular differentiation is being replaced with the notion of lineage plasticity (Box 1), as mounting evidence sugggests that cells possess the ability to change cellular identity from one developmental lineage to another. The change in cellular identity can be achieved through at least these three different routes: dedifferentiation, trans-determination and transdifferentiation [2,4,5]. Dedifferentiation is a process by which a differentiated cell type can shed its specialized function and acquire a stem/progenitor-like phenotype [6]. In trans-determination, a stem/committed progenitor cell becomes another [7,8]. On the other hand, during transdifferentiation, a differentiated cell of one lineage changes its identity and becomes a differentiated cell of another lineage with or without the involvement of an intermediate dedifferentiation step [4,9]. These processes are important for tissue repair and regeneration [10]. However, deregulated plasticity is also a key feature of cancer progression and therapeutic resistance [2,11]. Cancer cells are known to exploit their inherent potential to alter cell states as a mechanism to generate intratumoral diversity, to vanquish the constraints imposed, for instance, by the lack of oxygen/nutrients, the varying microenvironments encountered during metastasis and to overcome the drug-induced stress resulting in relapse.

Box 1Glossary: Definitions of cellular identity and plasticity.Cellular identity: The phenotypic and functional features that define the differentiation state of a cell.Cellular plasticity: Ability of the cells to dynamically and reversibly switch from one phenotypic state to another. (Although the terms cellular and lineage plasticity have sometimes been used interchangeably in the literature, lineage plasticity can be considered as a subcategory of cellular/phenotypic plasticity which involves a change in the differentiation state of the cell. The reversibility of lineage plasticity has also not been investigated thoroughly yet. Throughout this review, we have used cellular/lineage plasticity to denote the ability to convert from one differentiation state to another.)Histological transformation: The conversion of one cell type/lineage to another, identified based on histology.

## 2. Lineage Plasticity in Normal Development and Tissue Repair 

Evidence over the past several decades has shown that cellular identity is not always irreversibly fixed and can be altered stochastically or through experimental manipulation [12,13,14,15]. One of the earliest evidences for the plasticity of embryonic cells comes from the studies of Hans Driesch using sea urchin embryo [16]. He found that the early blastomeres till 2–4 cell stage maintained the potential to generate the entire organism, contradicting Roux’s view that the embryonic cell’s fate was determined after the first cell division itself [16]. Cellular plasticity is envisaged to be progressively restricted by epigenetic silencing of stemness-associated genes and/or genes associated with other lineages [11,17] (Box 1). The mature cell types thus generated are considered to be terminally differentiated in adults and a small pool of stem or progenitor cells is maintained in the adults through their life to replenish the lost or damaged cells. Although less apparent under normal physiological conditions, differentiated/mature cells display extensive plasticity and can dedifferentiate or change their identity to become another cell type in response to extreme stresses such as injury [18]. Extensive lineage plasticity following injury has been observed in different organs such as liver, pancreas, kidney, prostate and intestine [2,5,16,18]. 

An archetypal example of cellular plasticity contributing to regeneration following injury is seen in the liver [19]. Biliary epithelial cells can regenerate hepatocytes following chronic injury as demonstrated by Deng et al. using elegant lineage-tracing experiments [20]. Chemical-induced injury to biliary epithelial progenitor cells was shown to cause mature hepatocytes to convert into the progenitors. Interestingly, this conversion is reversible, and the cells can differentiate back to hepatocytes after removal of injury [21]. 

Another interesting example of lineage plasticity following tissue injury is witnessed in the pancreas. Near-total loss of insulin-producing β cells in adult mice caused the glucagon-producing α cells to undergo spontaneous reprogramming and transdifferentiate into β cells identified via lineage-tracing analysis. The transdifferentiated cells expressed both α and β specific markers [22]. Six months post-ablation, the mice no longer required exogenous insulin, indicating recovery from the injury. Interestingly, in juvenile mice, following the loss of β cells, no significant α cell conversion to β cell is seen; instead, β cells were generated by the spontaneous reprogramming of somatostatin-producing δ cells, thus suggesting an alternative mechanism of transdifferentiation [23]. Therefore, both α and δ cells could act as sources of β cells following injury, depending on the age of the mice. 

Cellular plasticity is also observed in the intestine following injury [5]. Lgr5^+^ stem cells located in the intestinal crypts give rise to enterocytes and secretory cells of the intestine. However, the ablation of Lgr5^+^ stem cells via diphtheria toxin-induced apoptosis causes another reserve cell population expressing Bmi1 (residing at a higher position in the intestinal crypts) to compensate for the loss of Lgr5^+^ cells. Similarly, irradiation-induced loss of Lgr5^+^ cells caused the Dll1^high^ cells, a committed progenitor population, to replenish the lost Lgr5^+^ cells, which in turn can generate the other cell types of the intestine [5,24].The phenomenon of injury-induced plasticity is also observed in other organs such as kidney, prostate, lung and salivary gland [25]. 

As exemplified by the accumulated data in the field of regenerative medicine, signals emerging from the damaged tissues post injury or stress can induce remarkable cell plasticity, resulting in dedifferentiation or a mature cell acquiring a new identity, which is not a part of the normal tissue homeostasis mechanism [26,27]. These studies emphasize the importance of exploiting cellular plasticity for therapeutic applications.

## 3. Cellular and Lineage Plasticity in Cancer

Although plasticity is a boon during tissue repair and wound healing, it becomes a curse when it is hijacked by cancer cells in their favor [27]. Unlike their normal counterparts where cellular plasticity is tightly regulated, the process can be aberrantly activated in cancer cells [28], potentially due to compromised genomic stability and rewiring of regulatory networks driving cell-fate commitment. Thus, deregulated plasticity can be implicated in cancer initiation, progression, metastasis and therapy resistance in diverse cancer types [2], which will be discussed briefly in the subsequent sections (Figure 1). 

### 3.1. CSC Plasticity

Many cancers, like normal tissues, were thought to be hierarchically organized, with a small fraction of cells called cancer stem cells (CSCs) residing at the apex of the hierarchy which is responsible for tumor growth, generating intratumor heterogeneity, metastasis and therapeutic resistance [29,30]. However, this framework has been replaced by the plasticity model which implies that the cancer stem cell state is not a defined entity but instead a dynamic state which can be gained or lost [30,31,32]. In other words, CSCs and non-CSCs can switch between each other [31]. Moreover, distinct subsets of CSCs can interconvert among one another [30]. CSC plasticity has been observed in different cancer types such as breast [30,33,34], glioblastoma [35,36] and melanoma [37,38]. This ability to switch between different phenotypic states can be regulated by both cell-intrinsic (genetic/epigenetic) and cell-extrinsic factors (microenvironment) [31,39]. For instance, a recent study found that glioblastoma cells expressing CSC-associated surface markers (CD133, A2B5, SSEA and CD15) do not represent a clonal entity but rather a dynamic state that changes with the varying microenvironment [36]. This ability to switch back and forth between stem-like and non-stem-like states is of critical importance in the clinical setting, as the previous idea of targeting the CSCs alone is not enough to curb cancer progression [30,40,41,42]. It further underscores the importance of understanding the molecular mechanisms underlying the plasticity for complete and effective eradication of all tumor cell populations.

### 3.2. Endothelial Trans-Differentiation and Vasculogenic Mimicry

A canonical mode of lineage plasticity is transdifferentiation in which one differentiated cell type becomes another [5,43]. For example, cancer cells have been shown to transdifferentiate into endothelial-like cells to promote neovascularization and tumor progression [44,45]. In a process known as vasculogenic mimicry (VM), tumor cells acquire vascular-like phenotype and form vascular channels on their own, independent of endothelial cells, and provide the nutrients and oxygen required to meet the growing nutritional and metabolic demands of the tumor [46]. As the name suggests, these vascular structures formed by the tumor cells are not true blood vessels but a mere functional mimic [47]. VM was initially identified by Manoitis in uveal melanoma as a novel neovascularization mechanism [48]. Since then, VM has been later reported in several other cancers [48,49,50,51,52], endorsing its role as a potentially important vascularization mode opted by tumors [53]. VM is associated with tumor progression, metastasis and poor survival [45,46,54]. Histologically, VM is identified based on a Periodic Acid Schiff (PAS) positive and CD31 negative staining, which distinguishes vascular mimetic blood vessels from angiogenic vessels [55]. 

The molecular mechanisms and pathways underlying the process of VM are still under investigation. The components in the tumor microenvironment such as cancer-associated fibroblasts (CAFs), tumor-associated macrophages (TAMs) have been found to regulate VM [46,56,57]. Hypoxia in the tumor microenvironment is known to induce VM and epithelial-to-endothelial transdifferentiation (EET) [44,58]. Moreover, non-coding RNAs such as long non-coding RNAs (lncRNA) and microRNAs (miRNA) are important in the VM process [59,60,61].

Tumor cells transdifferentiated into endothelial cells can again undergo another transdifferentiation process termed endothelial-to-mesenchymal transition (EndMT), which involves the conversion of endothelial cells into mesenchymal phenotype as observed in a recent study [62]. In vivo lineage tracing in melanoma mouse model has shown that metastatic melanoma cells hijacked the authentic vasculature and formed an intravascular niche at the metastatic organ [62]. These tumor cells transdifferentiated and became indistinguishable from endothelial cells (melanoma-endothelial transdifferentiation, EndT) and expressed endothelial markers such as CD31, VE-CAM. However, EndT observed in this study is different from VM, as cells involved in VM rarely express CD31 [63]. While VM is associated with primary and metastatic lesions, the endothelial transition observed in this study was observed intravascularly and not within the lesions [62]. These transdifferentiated tumor cells at small pulmonary vessels remained quiescent and were reawakened during metastatic seeding by endothelial-mesenchymal transition (EndMT).

### 3.3. Epithelial-Mesenchymal Plasticity

A process similar to EndMT is epithelial-mesenchymal transition (EMT): a reversible change in molecular, morphological and functional traits of epithelial cells to more mesenchymal traits during the metastatic cascade and/or in the emergence of drug resistance [62,64]. The process of EMT was first identified in embryonic development in higher chordates, where the mesenchymal tissue was observed to be formed by the primary epithelial cells through a cell-state transition [65,66]. EMT is a fundamental cell biological process during development, wound healing and also pathological conditions such as fibrosis and cancer [67]. It involves a reduction in epithelial traits such as cell–cell adhesion and apico-basal polarity and a concomitant gain of mesenchymal traits such as increased invasion and migration. 

Earlier thought of as a binary process, EMT is now viewed as a spectrum of states, including one or more hybrid epithelial/mesenchymal (E/M) states [68]. EMT is viewed as a fulcrum of cellular plasticity in carcinomas; moreover, sarcomas have been reported to undergo the reverse of EMT–MET (mesenchymal-to-epithelial transition) [69]. EMT is influenced by a multitude of pathways which include transforming growth factor β (TGFβ), Wnt–β-catenin, bone morphogenetic protein (BMP), Notch, Hedgehog and receptor tyrosine kinases [70]. These signaling pathways modulate the EMT transcription factors such as ZEB, TWIST and SNAIL which govern the levels of epithelial molecules such as E-cadherin and/or induce the expression of various mesenchymal ones [71,72]. ZEB and SNAIL form mutually inhibitory feedback loops with the two microRNA families, miR200 and miR34 [72,73,74]. Such loops can stabilize the epithelial and mesenchymal states. 

Recently, factors that can stabilize one or more hybrid E/M states have been identified too: GRHL2, OVOL2, NUMB, NRF2, NFATc and SLUG [75,76,77,78,79,80,81,82]. Hybrid E/M cells are more plastic when compared to cells that have undergone a complete EMT or MET [83,84], making them “fittest” for metastasis [85]. With the discovery of many intermediate states between canonical epithelial and mesenchymal states, EMT is being rechristened as EMP (epithelial-mesenchymal plasticity) [68]. 

While the association of EMP with lineage plasticity requires further investigation, we hereby discuss the cases which seem to tie EMP most closely with lineage plasticity: carcinosarcomas. Carcinosarcomas (CS) are cancers which possess both carcinomatous and sarcomatous features but arise from a single progenitor [86]. In a few cases, the sarcomatous component is said to arise from the carcinomatous components through the process of EMT [87]. Carcinosarcomas have been observed mostly in female reproductive organs—uterus and ovaries. 

Uterine carcinosarcomas (UCS) were initially called mixed Mullerian tumors. They are termed to be metaplastic as they arise from a single epithelial cell [88]. EMT in UCS is modulated by anaplastic lymphoma kinase (ALK) [89] by activating TWIST. UCS often bears Trp53 mutations, thereby disrupting the p53, PI3K and Fbxw7 pathways. The loss of function of Fbxw7 gene was found to be a major driver of EMT, facilitating the formation of UCS from an endometrial epithelial cell [90]. Fbxw7 alters the EMT axis by degrading ZEB in a GSK-3β phosphorylation-dependent manner [91]; consistently, overexpression of Fbxw7 was shown to downregulate TWIST in UCS [90]. Further, the expression of mutant histones (H2A and H2B) has also been shown to be associated with increased EMT marker levels and enhanced migration and invasion, ultimately transforming uterine serous carcinoma to UCS [92]. Whether this transformation is attributed to lineage plasticity remains to be investigated. 

In UCS patient tissue samples too, EMT has been observed. When categorized into carcinomatous, transitional and sarcomatous regions, it was observed that the sarcomatous region of tissues showed higher EMT expression markers, namely ZEB1 and SNAI2, as compared to the carcinomatous region. Moreover, the level of ZEB1 was higher in the sarcomatous region when compared to the transitional region [93]. Similar to UCS, ovarian carcinosarcoma (OCS) showed enriched expression of EMT markers when compared to the cohort of high-grade serous carcinoma in TCGA [87], suggesting that OCS indicates phenotypic and/or lineage plasticity signatures through the EMT landscape. 

Such biphasic carcinosarcomas are very rare but highly aggressive cancers. Patients show poor prognosis even when the disease is detected at very early stages [94]. Even though there is emerging evidence to say that the carcinosarcoma cells arise from a common progenitor (epithelial/carcinomatous cell) through the process of EMT, the nature and mechanism of these transitions still remain poorly understood. Further analysis of in vitro, in vivo and ex vivo CS models can accelerate our understanding of lineage and cellular plasticity mechanisms in CS and consequent therapeutic targets. Recent observations have shown that targeting TGFβ pathway in UCS with galunisertib (GLT) showed decreased viability and invasiveness [95].

### 3.4. Lineage Plasticity in Leukemia

Leukemias can also exhibit manifestations of lineage plasticity. Leukemia is the umbrella term used to define the cancers of hematopoietic cells, namely the lymphocytes (lymphocytic) and that of the bone marrow (myelogenous). Leukemia can be both acute and chronic. Lymphocytic leukemia (CLL) is a disease characterized by the continual accumulation of B lymphocytes (CD5+ cells) in the bone marrow, peripheral blood lymph nodes and spleen (secondary lymphoid organs) [96]. On the other hand, myeloid leukemia involves the clonal expansion of myeloid cells that are transformed and are primitive hematopoietic progenitor cells [97,98]. 

In certain cases, co-existence of markers indicating both lymphoid- and myeloid-lineage markers, or T-cell and B-cell markers have been observed [99]. In leukemia, a bidirectional switch has been observed between lymphoblastic leukemic cells and myeloblastic leukemic cells [100]. Within lymphoblastic leukemia, a switch has been observed between T-cell acute lymphoblastic leukemia (ALL) and B-cell ALL. The patient initially was diagnosed with T-cell ALL but after remission, flow cytometry data showed enrichment of B-lymphocytes [101]. The switch in lineage during relapse suggests malignancy is transferred between lineages. Similar observations were seen in mice carrying a heterozygous loss of Pax5 and Ebf1, which are essential for stable B-cell lineage commitment. In these mice, when Notch signaling was activated, it resulted in a lineage switch accompanied by an expansion of CD19-negative leukemia cells [102]. In rare cases, T-cell ALL can switch to acute myeloblastic leukemia (AML). In a patient diagnosed with T-ALL, relapse was seen after 50 days of treatment with standard T-ALL chemotherapy; upon relapse, AML signatures were seen. This patient also showed a switch back from AML to T-ALL when subjected to bone marrow transplant followed by second round of therapy [103]. This switch from T-ALL to AML has also been observed in vitro and also in vivo mouse models. Leukemia produced using CD4/CD8 double-negative (DN2) T-cell progenitors expressing either Myc/ Bcl2 or MLL-AF9 oncogenes comprised three phenotypic populations, i.e., myeloid, T-cell and bi-phenotypic in the same recipient. The clonal relationship and hence a lineage switch between the three fractions were established based on T-cell receptor rearrangement [104]. 

Put together, in vitro and in vivo observations and clinical reports both support the existence of different instances of lineage plasticity in leukemic cells. Further mechanistic understanding of the relative frequency of such possible transdifferentiation events and necessary and sufficient conditions required for such events needs to be mapped out mechanistically. 

### 3.5. Lineage Plasticity and Therapeutic Resistance

Lineage plasticity can be a driver of therapeutic resistance or a consequence of therapy-induced stress. In many cancers, to circumvent therapy-induced stress, tumor cells can dedifferentiate into a stem-like state or differentiate into alternative cell types [2,105,106]. Drug treatment could also help select for rare cells that are already tolerant which could survive the treatment and get reprogrammed to acquire resistance [107]. The most widely studied manifestation of lineage plasticity contributing to therapeutic resistance involves the change in histological subtype as commonly seen in lung, prostate and pancreatic cancers [108,109,110]. 

Malignancy in prostate cancer is driven by androgen signaling. Traditionally, treatment for prostate cancer involves androgen deprivation or antagonization of androgen receptors. However, chronic androgen pathway depletion leads to the emergence of resistance and eventually leads to castration-resistant prostate cancer (CRPC). Amongst the multiple mechanisms adopted by prostate cancer cells to achieve therapeutic resistance is the treatment-induced lineage crisis [111] and consequent differentiation of prostate adenocarcinoma cells into neuroendocrine (NE) lineage leading to neuroendocrine prostate cancer (NEPC). The degree of neuroendocrine differentiation increases with tumor progression and is correlated with poor prognosis in patients [110,112]. Whole exome sequencing of metastatic prostate cancer biopsies showed that the treatment-resistant neuroendocrine variant can arise from the adenocarcinoma through divergent clonal evolution [113]. Neuroendocrine reprogramming is characterized by the acquisition of neuroendocrine-specific markers like chromogranin A and synaptophysin [114,115]. Lineage tracing showed that these neuroendocrine cells arise from luminal adenocarcinoma cells through transdifferentiation [108]. Multiple genetic and epigenetic mechanisms have been identified in mediating this phenotype switch, including the genomic loss of TP53, RB, PTEN and gain of MYCN and AURKA [116]. Epigenetic modulators as well as lineage-determining transcription factors as such were shown to promote plasticity towards the neuroendocrine phenotype [108,117,118,119,120,121]. A recent study has shown that Mucin 1 (MUC1), a transmembrane glycoprotein, was found to promote androgen independence and self-renewal of prostate cancer cells [122]. MUC1 was shown to induce BRN2, an NEPC-associated transcription factor, through MYC-mediated mechanism, and silencing MUC1 caused suppression of BRN2.This study highlights the significance of MUC1 in prostate cancer lineage plasticity, making it an attractive target for therapeutic inter-vention. Another interesting study has revealed the role of Tribbles 2 (Trib2), a pseudokinase, in mediating antiandrogen resistance in prostate cancer. They found that antiandrogen treatment-resistant cells overexpress Trib2 which can induce lineage switching by upregulation of SOX2 and BRN2 [123].

Similarly, therapy resistance due to histologic transformation has also been observed in lung carcinoma. Lung cancer is a heterogeneous disease which can be broadly classified as small cell lung carcinoma (SCLC) and non-small cell lung carcinoma (NSCLC). SCLC comprises about 15–20% of lung cancers [109,124] and arises from pulmonary neuroendocrine cells with frequent alterations in TP53, RB and PTEN genes [124,125,126,127,128]. These tumors are not surgically resectable, as they are presented in late stage and are usually treated with a combination of chemotherapy and immunotherapy [109]. NSCLC, on the other hand, comprises 80–85% of the lung cancers; lung adenocarcinoma (LUAD) is the predominant subtype of NSCLC and arises from type 2 alveolar (AT2) cells [129]. EGFR tyrosine kinase inhibitors (EGFR TKIs) have been used for targeted therapy and have enhanced survival of advanced NSCLC patients, who failed to respond to chemotherapy [130]. However, patients eventually develop resistance to EGFR TKIs through several mechanisms, including histologic transformation to SCLC, which constitutes about 14% of the cases showing EGFR TKI resistance [131]. However, the transformation to SCLC was only observed in EGFR mutant LUAD and not in EGFR wild type LUAD [109]. Due to the lack of preclinical models that can faithfully recapitulate this transformation, much of our understanding of NSCLC LUAD to SCLC comes from the studies of clinical samples where patients initially diagnosed with LUAD and treated with EGFR TKIs later relapsed with SCLC tumor [109]. In addition to switching to SCLC-like, transition of LUAD cells to squamous cell carcinoma (SCC) accounts for a small fraction of EGFR mutant lung cancer patients leading to EGFR TKI resistance [132,133,134]. In addition, EMT also contributes to EGFR TKI resistance [109,135,136,137]. In SCLC patients, a reverse transition from neuroendocrine to a chemoresistant non-neuroendocrine phenotype has also been observed, which is mediated by Notch signaling [138]. 

Lineage plasticity has been reported to contribute to therapeutic resistance in basal cell, colorectal and hepatocellular carcinoma (HCC) as well [139,140]. Metastatic colorectal cancer (mCRC) patients who are chemorefractory with inoperable tumors are often treated with monoclonal antibodies targeting EGFR such as cetuximab and panitumumab. However, treatment response and survival benefit are suboptimal even in patients responsive to EGFR blockade therapy due to the emergence of drug tolerance. Paired samples of untreated and cetuximab-treated patient-derived xenografts showed that EGFR inhibition resulted in residual treatment-tolerant slow cycling cells with hyperactive Wnt signaling and Paneth cell-like differentiation, reminiscent of drug-tolerant persisters reported across contexts [141]. Similar Paneth cell-like differentiation has been observed in normal mouse intestine following EGFR blockade [142,143,144] and is generally absent in human and rodent colon [139]. Thus, cetuximab treatment activates Paneth cell-like lineage reprogramming that results in drug tolerance in mCRC, thus underscoring the role of lineage-adapted reprogramming as a mechanism adopted by the cancer cells to evade therapy. 

Sorafenib, a multi-kinase inhibitor, is the most widely used drug for first line treatment of advanced HCC [140,145]. However, its overall survival benefit is very low [140]. In a recent study, Claudin 6 (CLDN6), a tight junction transmembrane protein, was found to be associated with lineage plasticity, which correlates with sorafenib resistance [140]. CLDN6 is highly expressed in embryonic cells but is significantly downregulated in mature hepatocytes. Its expression was found to be upregulated in HCC patient tumor samples and correlated with poor survival. CLDN6 overexpression resulted in transdifferentiation of hepatocytes into biliary lineage as indicated by the marker expression and this change in identity was found to contribute to resistance to sorafenib. 

In basal cell carcinoma (BCC), Hedgehog signaling pathway inhibitor vismodegib has been used to treat patients with advanced and metastatic disease. Despite its efficacy, residual disease persists. Using a mouse model of BCC, Biehs et al. found that these drug-persistent cells were characterized by a permissive chromatin state, Wnt pathway activation, which enabled them to assume an alternate cellular identity that no longer relied on the original driver-Hh signaling for survival, enabling them to evade the treatment [146].

Targeted therapy has also been attributed to the phenomenon of lineage switching in leukemia. Blinatumomab, a monoclonal antibody, targets both CD19 on B cells and CD3 on cytotoxic T cells, facilitating the lysis of the B cell by the host cytotoxic T cells. Upon hematopoietic stem cell transplant followed by treatment in a patient, the remission indicated the emergence of acute myeloid lymphoblasts with no evidence of residual B-ALL blasts other than the newly formed myeloid blasts [147]. Similarly, in a chronic lymphocytic leukemia (CLL) patient, treatment with PI3Kδ inhibitor led to a switch from lymphocytic leukemia to Langerhans cell histiocytosis (LCH) with acquired BRAF V600E and STK11 mutations, and loss of expression of B-cell lineage markers such as PAX-5 [148]. Another instance of therapy-driven lineage switch was observed in a patient with a twelve-year history of mantle cell lymphoma (MCL). Upon administration of autologous chimeric-antigen receptor T-cells targeting CD19 (CART19) therapy, a switch from MCL into poorly differentiated sarcoma was observed [149]. Interestingly, Jacoby et al. [150] observed that in pre-B cell ALL in in vivo murine models, persistent CD19 CART cell therapy resulted in a myeloid lineage switch characterized by the loss of CD19. Further analyses revealed that the resistant clones emerged as a consequence of genetic reprogramming. This study established lineage switching as a mechanism of treatment resistance in leukemia. 

Targeted therapy can also drive the switch from myeloid to lymphocytic lineage. For instance, imatinib treatment for a CML patient revealed B-lymphoblastic leukemia through bone marrow study and immunophenotyping by flow cytometry [151]. In a meta-analysis of 726 children who were diagnosed with B-cell precursor ALL, 8% of them exhibited a switch from ALL to monocytic lineage accompanied by loss of B-cell immunophenotype, including CD19 expression [152]. In another study, lineage switching upon treatment of two B-ALL patients with CAR T-cells was recorded. In an 18-year-old male patient with CD19+ B-ALL, eight months post-treatment biopsy of extra-orbital soft tissue mass formed revealed myeloid sarcoma with no immunophenotypic evidence of B-ALL. Another 19-year-old female with B-ALL demonstrated a switch from B-ALL to ambiguous lineage T/myeloid acute leukemia post-treatment [153]. Similar behavior has been reported in pediatric leukemia as well [154]. 

Together, all these observations across multiple cancers emphasize the necessity of better understanding the enigmatic process of lineage plasticity for therapeutic success. 

## 4. Mechanisms Regulating Lineage Plasticity in Cancer

Various tumor cell intrinsic and extrinsic factors have been demonstrated to be involved in regulating lineage plasticity. Loss of tumor suppressor genes such as RB1, PTEN and TP53 has been associated with the acquisition of lineage plasticity in multiple cancers [155]. Moreover, combined loss of Trp53 and Rb1 can lead to plasticity and transdifferentiation in both prostate and lung cancer as shown in mouse models [156,157]. 

Ever since the discovery of the four reprogramming factors (Oct4, Sox2, Klf4 and c-Myc) by Takahashi and Yamanaka, many other transcription factors have also been identified that contribute to lineage reprogramming in development. OCT4, a homeodomain transcription factor, is a critical factor involved in mammalian early embryonic development and cancer progression. It is considered as a CSC marker for multiple cancers [158]. In head and neck squamous cell carcinoma, Oct4-high cells tend to display more stem cell-like characteristics such as self-renewal, chemoresistance and invasion capacity compared to Oct4-low ones [159]. A recent study on liver cancer explored the role of Oct4 in tumor vasculogenesis and revealed the differential role of two Oct4 variants, Oct4A and Oct4B1, in mediating endothelial reprogramming of liver cancer stem cells (LCSCs) into tumor endothelial cells (TECs) in vitro [160]. Accumulating evidence suggests that Sox family members are upregulated in breast cancer and are involved in promoting tumor progression, invasion, metastasis and chemoresistance. Recently, SOX9 was identified as an important regulator of luminal to basal reprogramming in basal-like breast cancer [28]. Despite their basal features, BLBC cells are likely to have a luminal progenitor origin as evident from various previous studies [161,162,163]). Sox9 was found to be a determinant of estrogen receptor negative (ER-) luminal stem/progenitor cell (LSPC) activity and this is mediated in part by upregulation of non-canonical nuclear factor κB (NF-κB) signaling pathway [28]. Inactivation of BLBC tumor suppressors TP53 and RB by SV40Tag causes the upregulation of Sox9 in ER- LPSCs and consequently mediates the luminal-to-basal lineage reprogramming in vivo. Moreover, deletion of Sox9 inhibits the progression of ductal-like lesions to invasive carcinoma, implying the crucial role of Sox9 as a driver of BLBC lineage reprogramming and tumor progression. SOX2, another SOX transcription factor, was found to promote plasticity and antiandrogen resistance in TP53 and RB1-deficient prostate cancer using in vitro and in vivo human prostate cancer models [119]. In lung cancer, an epigenetic switch between SOX2 and SOX9 confers phenotypic and oncogenic plasticity, enabling the cells to alter between proliferative and invasive states [164]. MYC is another important transcription factor upregulated in different cancers [165]. N-MYC and c-MYC are well-studied drivers of neuroendocrine plasticity in prostate cancer [117,166]. In pancreatic ductal adenocarcinoma (PDAC), c-MYC mediates ductal-to-neuroendocrine plasticity and gemcitabine response which in turn leads to poor outcome and therapeutic resistance [110]. 

Another important family of transcription factors, the zinc-finger E-box-binding homeobox family of transcription factors (ZEBs), ZEB1 and ZEB2, is known for its ability to induce EMT-driven cellular plasticity in carcinoma progression [167,168]. However, in melanoma, ZEB1 and ZEB2 function antagonistically and play a central role in “phenotype switching” by modulating the expression of microphthalmia-associated transcription factor (MITF), a master regulator of melanocyte homeostasis and differentiation. Increased ZEB2 expression drives the expression of MITF, leading to a differentiated and proliferative cell state whereas increased ZEB1 expression lowers MITF and leads to a stem-like and invasive cell state [167,169,170,171].

Although most of the initial studies focused on genetic alterations, the involvement of epigenetic determinants in modulating lineage plasticity is now being increasingly appreciated [11,39,172,173,174]. Mounting evidence suggests that epigenetic modifications including DNA methylation, histone modification and chromatin remodeling play a key role in promoting lineage plasticity in tumor progression [2,11,39,175,176,177]. For example, in prostate cancer, N-Myc was found to induce enhancer of zeste homologue 2 (EZH2)-mediated transcription program [117]. EZH2 is a component of the polycomb repressor complex (PRC) that silences target gene through trimethylation of lysine 27 of histone 3 (H3K27) [178]. EZH2 interacts and cooperates with N-Myc to reduce the expression of N-Myc target genes and drives neuroendocrine plasticity in prostate cancer [117]. EZH2 has also been implicated in controlling the molecular subtype identity and plasticity in PDAC [179]. Combined analysis of RNA and chromatin immunoprecipitation sequencing revealed that EZH2 causes progression of PDAC from a less aggressive “classical” subtype to a more dedifferentiated, aggressive and therapy-resistant “basal” subtype by repression of a classical subtype-associated transcription factor, GATA6. Depletion of EZH2 and re-expression of GATA6 resulted in a subtype switch from the aggressive to classical subtype, thus highlighting EZH2 as a promising therapeutic target in PDAC [179]. Besides histone methyl transferases, histone demethylases have also been reported in lineage plasticity, especially in NSCLC and glioblastoma [2,180,181]. Furthermore, chromatin remodelers are also critical regulators of lineage plasticity, such as the mammalian switch sucrose non-fermenting (mSWI/SNF) complex also known as Brg/Brahma-associated factors (BAF) [182]. In a recent study in NEPC [118] using cancer cell lines, patient-derived organoid and large patient datasets, the expression of SWI/SNF complex was found to be altered in NEPC and higher expression of SMARCA4 (BRG1), an SWI/SNF subunit, correlated with disease aggressiveness. Neuroendocrine transdifferentiation was found to be a result of the differential interaction of the SWI/SNF complexes with distinct lineage-specific factors in castration-resistant prostate cancer compared to prostate adenocarcinoma. SWI/SNF complex is also associated with plasticity in estrogen receptor positive (ER^+^) breast cancer. ARID1A, a subunit of SWI/SNF complex, was recently identified as a determinant of luminal lineage identity in ER+ breast cancer. Loss of ARID1A expression in tumor cells and patients determines resistance to endocrine therapy by promoting a switch from ER-dependent luminal cells to ER-independent basal cells [183]. Besides epigenetic factors, long non-coding RNAs can also modulate cellular plasticity as shown in lung and prostate cancer [184,185,186,187]). Genetic and epigenetic modulators can crosstalk and promote cellular plasticity; for instance, in prostate cancer, TP53 and RB1 loss can cause upregulation of SOX2 and EZH2 [156,188] and consequently an epigenetically permissive state facilitating lineage plasticity. 

In addition to genetic and epigenetic mechanisms, cell-extrinsic factors such as inflammation, microenvironment and therapeutic stress can induce cellular plasticity [105]. Chronic inflammation has long been identified as a hallmark of cancer [189] and is an important player in tumor progression and metastasis. One of the best examples of inflammation-induced lineage plasticity is in metaplasia or the replacement of one differentiated cell type by another differentiated cell type in the same tissue [190]. In normal tissues, injury and chronic inflammation can induce metaplasia and change in cellular identity. Transdifferentiation is one of the mechanisms by which metaplasia can arise in tissues [191,192]. Although considered a protective mechanism against damage, metaplasia is associated with an increased predisposition to cancer [191]. Inflammation is also associated with prostate cancer initiation by mediating a basal-to-luminal differentiation [18]. This inflammation-induced luminal differentiation and cancer initiation was found to be augmented by the loss of a homeobox gene NKX3.1 [193]. Inflammatory cytokines, proinflammatory cytokines and inflammation-associated myeloid cells are the three key inflammatory axes associated with stemness and EMT in breast cancer plasticity and malignancy [194,195]. Different components in the tumor microenvironment (TME) such as fibroblasts, macrophages, endothelial cells and infiltrating immune cells can conspire with the tumor cells to promote tumor cell plasticity [11,196]. In breast cancer, for example, cancer-associated fibroblasts (CAFs) were found to determine the molecular subtype of breast cancer by engaging in paracrine crosstalk with tumor cells via platelet-derived growth factor-CC signaling [197]. Disrupting the PDGF-CC signaling through genetic or pharmacological intervention in mouse models resulted in the conversion of basal breast cancer cells into ER+ luminal cells, making them susceptible to endocrine therapy [197]. Tumor-associated macrophages (TAMs) are also a major component in TME involved in facilitating tumor progression and metastases [198]. For instance, in prostate cancer, TAM-derived CCL5 promotes EMT, stemness and metastases by activating β-catenin/STAT3 signaling [199]. Another study has shown that inhibiting growth arrest-specific protein 6 (Gas6), a multifunctional protein secreted by TAMs and CAFs, resulted in inhibiting EMT and metastases in pancreatic cancer [200]. 

In addition to the stromal cells, components of the extracellular matrix (ECM) can also instigate cellular plasticity [201]. Matrix stiffness can promote EMT and stemness in breast, colorectal and liver cancers [202,203,204]. Extracellular proteases can also stimulate cellular plasticity [201,205,206]. In melanoma and glioblastoma, A disintegrin and metalloprotease with thrombospondin motifs 1 (ADAMTS1), a multifunctional metalloproteinase, is linked to endothelial transdifferentiation [205,207]. Thus, it is evident that there are multiple mechanisms involving cell intrinsic, epigenetic as well as microenvironmental factors that contribute to lineage plasticity in different cancers (Figure 2). A key question is to delineate the interplay between these mechanisms, which adds an extra layer of complexity to the process of shifting between lineages in normal development and cancer.

## 5. Therapeutic Targeting of Lineage Plasticity—Taming the Shape Shifter

It is becoming increasingly clear that lineage plasticity is a major driver of tumor progression and therapeutic resistance, making it imperative to identify the crucial molecular mediators underlying the process which can be exploited for therapeutic targeting. Multiple strategies can be used for targeting lineage plasticity as exemplified by recent studies. One way this could be achieved is by directly inhibiting the mediators of lineage plasticity. These mediators include chromatin-modifying enzymes (histone deacetylases (HDACs)) [208], histone demethylases KDM6A/B [181], signaling pathways (Wnt) [146], IL6-STAT3 [209,210], etc. Since TME is also a critical regulator of cellular plasticity, targeting the TME-derived components along with the cancer cells is an attractive strategy to combat the cancer plasticity [197,211,212]. 

The second approach is the direct targeting of the new cell fate acquired through lineage switching [140,211]. For example, in advanced HCC, CLND6 overexpression has been associated with lineage plasticity and treatment resistance. HCC cells with CLDN6 overexpression could be eliminated by using cytotoxic drug (DM1) conjugated monoclonal antibody against CLDN6 [140]. The reversal of the lineage plasticity is another plausible therapeutic option [107]. In NEPC, inhibition of EZH2 resulted in the reversal of lineage switching and restored cells’ sensitivity towards enzalutamide treatment [107,117]. Moreover, by blocking TGFβ signaling using forskolin and cholera toxin, in basal-like breast cancer cells, EMT could be reverted to MET, which increases sensitivity towards anti-cancer therapy [211,213]. Another interesting approach is to harness the intrinsic pliability of cells to assume a new identity and leverage this malleability for therapeutic targeting. This is evident from the earlier studies on acute promyelocytic leukemia (APL) where differentiation therapy using all-trans-retinoic acid (ATRA) causes the terminal differentiation of APL blasts into granulocytic cells, thereby enabling good treatment response and cure [214]. Similar strategy has also been applied in solid cancers [191,215,216]. A recent study explored the possibility of exploiting the cellular plasticity of cancer cells undergoing EMT for therapeutic targeting of cancer. Using MEK inhibitor in combination with rosiglitazone (an adipogenesis inducer), they showed that breast cancer cells undergoing EMT can be induced to transdifferentiate into post-mitotic adipocytes which lack cellular plasticity [217]. Moreover, combining traditional chemotherapy or targeted therapy along with such transdifferentiation approaches could hold great potential as a promising treatment for cancer [218]. These studies shine a ray of hope for the treatment of cancer patients by targeting the inherent cellular plasticity of cancer cells (Figure 3). 

The advent of single cell sequencing techniques has been instrumental in shaping our knowledge regarding the molecular basis of cell state transitions and cellular plasticity in an unprecedented fashion [219,220]. The insights gained from such analyses and mechanism-based mathematical and statistical models can be utilized for the development of better therapeutic strategies [221,222]. Considering the enormous significance of plasticity in normal tissue homeostasis and regeneration, targeting cellular plasticity is like unlocking the Pandora’s Box and therefore warrants greater understanding of the process before we tinker with it.

## Figures and Tables

**Figure 1 cancers-13-03602-f001:**
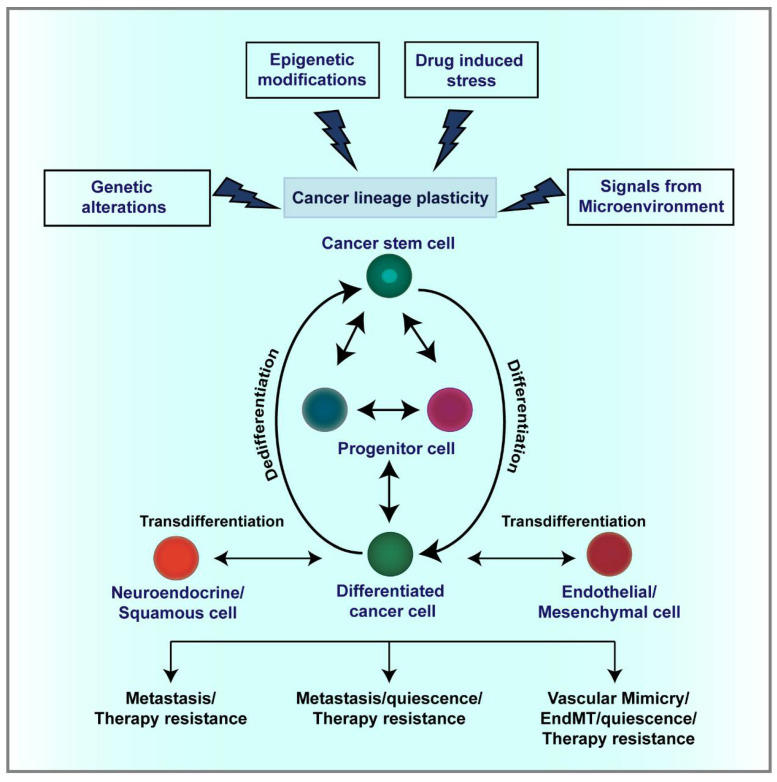
Lineage plasticity promotes tumor progression, therapy resistance and metastasis. Cancer cells can dynamically alter their identity as a result of genetic/epigenetic alterations or in order to adapt to the drug-induced stress and varying microenvironmental cues. This ability, termed as lineage plasticity, enables them to switch between different cell states through processes such as dedifferentiation or transdifferentiation. During dedifferentiation, differentiated cancer cells become less specialized and acquire stem/progenitor cell-like characteristics. Through transdifferentiation, cancer cells can convert from one differentiated lineage to another, i.e., from luminal adenocarcinoma cells to neuroendocrine/small cell or squamous cells as observed in lung, prostate and pancreatic cancer. Cancer cells have also been found to adopt endothelial cell characteristics to form vascular channels through a process called vasculogenic mimicry or mesenchymal phenotype through the process of epithelial-mesenchymal transition. This lineage plasticity is increasingly being recognized as a key player in quiescence, therapeutic resistance and metastasis in cancer.

**Figure 2 cancers-13-03602-f002:**
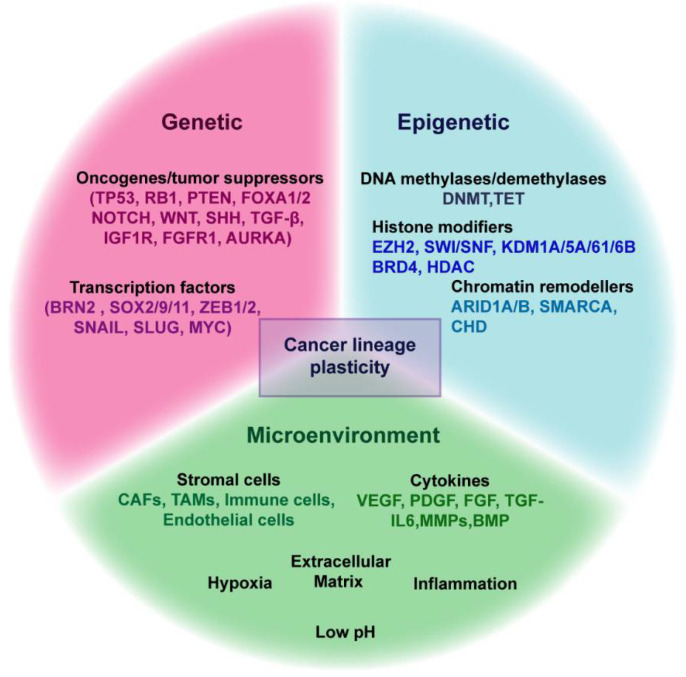
Genetic, epigenetic and microenvironmental factors involved in regulating lineage plasticity. The genetic factors include the loss of tumor suppressors and gain of oncogenes and overexpression of several transcription factors, eventually resulting in the enabling of the cellular reprogramming. In addition, altered expression of epigenetic modulators and microenvironmental components also play an important role in facilitating the switching of cellular identity in response to various stresses.

**Figure 3 cancers-13-03602-f003:**
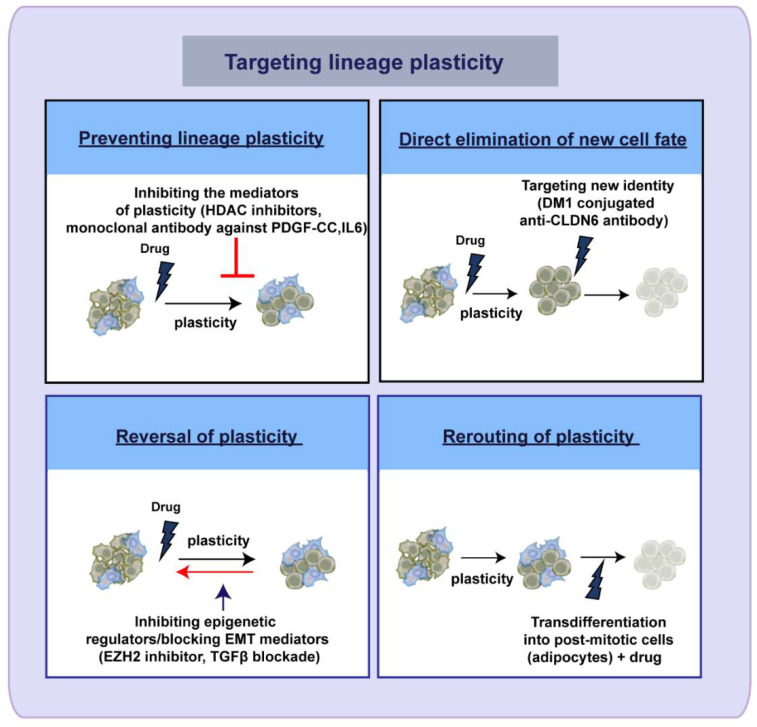
Therapeutic strategies to target the lineage plasticity. Schematic showing the approaches used to target cancer lineage plasticity. Lineage plasticity can be prevented by directly inhibiting the mediators regulating cellular plasticity (by inhibiting cancer cell-associated proteins (HDACs KDM, IL6-STAT3 signaling) and microenvironment-derived factors (PDGF-CC)). Another approach is to eliminate the new cell fate adopted by the cells through lineage switching (e.g., using cytotoxic drug-conjugated antibody against CLDN6). The third approach is to reverse the lineage plasticity of cancer cells (e.g., by inhibiting the epigenetic modulators such as EZH2 or by blocking EMT mediators like TGFβ). Another strategy that has recently gained attention is to exploit the intrinsic cellular plasticity of the tumor cells by transdifferentiating them into alternate cell lineages that are post-mitotic and thereby rerouting the lineage plasticity. Combining chemo/targeted therapy along with the transdifferentiating drug will facilitate the effective eradication of tumor cells.

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
