# Peer review of "Lineage Plasticity in Cancer: The Tale of a Skin-Walker"

_cancers, 2021, doi:10.3390/cancers13143602_

Round 1

Reviewer 1 Report

Review on manuscript by Thankamony et al. “Lineage plasticity in cancer: The tale of a skin-walker”

Overall, this review provides a comprehensive description of the existing literature and remaining questions related to lineage plasticity in cancer. Given the growing evidence on the prevalence, role and promise of targeting lineage plasticity in cancer, this work offers a significant contribution to the field. There are some moderate changes I would recommend making, related to clarifications, wording and perhaps rearranging certain sections for improving the manuscript.

Comments

  • Not specific to this review alone, but in general, there seems to be lack of clear definition between cell plasticity and lineage plasticity (which is linked to developmental processes) in the field. Although many times these terms are used interchangeably, they don’t always mean the same thing, providing an opportunity here, to perhaps better clarify and discuss. For example, cellular plasticity may also mean switching into a slightly different “phenotypic” state (e.g. one epithelial state to a partial EMT state, or a metabolic state to another, see Kondo et al., Cell Reports, 2021). Although phenotypic changes are also regarded as examples of cellular plasticity, they don’t represent an actual lineage change that is observed for example in leukemias, or in EndT. In relation to this, I wonder if section 7 should be titled Lineage Plasticity in Leukemia vs Cellular Plasticity to in general maintain consistency. Given that this is a comprehensive review of lineage plasticity, this should be briefly discussed since it is something that occurs throughout the whole paper (see also comment #5).

  • In the abstract it is presented that the authors will discuss lineage plasticity in solid tumors and leukemias. However, the solid tumors are discussed under section 8 “Lineage Plasticity and Therapeutic Resistance”, yet the literature related to leukemia is discussed under its own leukemia specific title. For better flow, maybe the authors should reconsider how they outline/title their sections. It may be better to break the sections into lineage plasticity in leukemia, solid tumors and therapy resistance, or keep them all as subsections under one section. The other reason I am suggesting this is because the literature discussed about leukemias, also relates (directly or indirectly) to therapy resistance/remission. For e.g. in section 7 it is implied that lineage switching happens due to therapy, but not presented as therapy resistance, whereas for similar phenomena in solid tumors (section 8) these are presented as therapy resistance. For example, in Jacoby et al., Nat Comm. 2016, lineage switch is presented as a mechanism of resistance to CAR therapy. To be fair, in the fields of Prostate Cancer and Lung Cancer, lineage transformation is presented as both a result and driver of therapy resistance, whereas in most leukemia studies, lineage transformation is presented only as a result of therapy, and an observation at time of remission. Both can be true, therapy may drive lineage switching, and the transformed cells can be resistant to therapy. This needs to be better clarified and presented at least as a possibility. There is also the possibility that therapy helps select for small populations that may be already “switched” and this is a different scenario that is not discussed. For example, in the field of NSCLC to SCLC transformation, this is always presented as a possibility – although admittedly it’s not the mainstream hypothesis (see review by Oser MG et al., Lancet Oncol., 2015).

  • Continuing on section 7, I believe the reference mentioned for lines 251-254 (#103) is misplaced. The paper specifically describes mantle cell lymphoma (MCL) transdifferentiation to poorly differentiated sarcoma (Sarc). It is however correctly mentioned in lines 269-271, but with a wrong description. In the paper by Zhang et al., it states that it is unlikely that CART19 therapy triggered transdifferentiation, and that resistance to CART19 was mainly based on loss of CD19. The patient was undergoing other types of therapy as well and they attribute transdifferentiation to those instead. (The same paper however does provide references that show cell-lineage perturbation leading to loss of CD19 as well).

  • Also related to the theme of resistance, in line 259, I believe the authors meant that lineage switching is attributed to targeted therapy and not the other way around. I am also not sure why in the paragraph starting in line 272, there is the introduction of the term “treatment-induced stress” when an example of imatinib is given (which is technically also an example of tyrosine kinase targeted therapy).

  • Are CSC plasticity and EMP considered/introduced as “types/examples” of cancer lineage plasticity, or processes related to lineage plasticity? This goes back to the initial comment that cellular plasticity and lineage plasticity are not necessarily the same. This needs to be more well described/clarified because there is literature that differentiates these. For example, EMT has been described/observed in a plethora of NSCLC studies; however, NSCLC to SCLC transformation is a rare phenomenon that is not equated to EMT per say but discussed as having “shared biological underpinnings” (See paper by Meador & Hata, Pharmacology & Therapeutics, 2020). In other cases, EMT is presented as a distinct mechanism of resistance of NSCLC to targeted therapy from that of histologic transformation (Leonetti et al., BJC, 2019). The same can be said in the section where authors discuss the role of the TME on plasticity (line 469 to the end of the section). Many of the examples provided are specific to EMT, so it all depends on if the authors are equating EMT with lineage plasticity or presenting it as an example of lineage plasticity.

  • In section 6 (EMP), Twist is omitted when EMP specific TFs are first introduced (lines 182-186) but then it is mentioned for the first time in line 202.

  • Focusing on lineage differences, it would be good to include that NSCLC cells derive from Alveolar type II cells (vs SCLC, where cell origins are attributed to neuroendocrine cells – as described in the text).

  • It would be best to include “in cancer” in the title of section 9 (mechanisms). I am not sure if lines 386-388 are needed, since they are relevant (pretty much the same) to lines 384-385.

  • For section 10, lines 513-514, be more specific/ give an example on what mediators of plasticity are (e.g. are these different from the epigenetic modulators in line 516?)

  • Figure 1. Legend 1 should be a bit more expanded for readers to understand more from the figure alone. Also, where Neuroendocrine/Squamous is placed, it would be good to include small-cell lung cancer transformation as well, and not only squamous. Finally, since lineage plasticity can also be drug-induced, perhaps it can be included in some way in the figure as an additional driver.

  • Figure 3. This figure can be a bit confusing (perhaps working on the legend and text may help with this). For e.g. shouldn’t all boxes have the first (or only arrow) in combination (or in sequence) with a drug? Throughout the paper, lineage plasticity is discussed in relation to established cancer therapies. Are the authors suggesting only targeting plasticity as therapy in the box to the left? How are the mediators of plasticity in the box to the left, different from the epigenetic regulators in the box to the right? Why in the box to the right are we first inhibiting plasticity and then promoting it again with transdifferentiation and not promoting from the start transdifferentiation, to lead the cells into a targetable state? In the box in the middle, it would make more sense to have the “targeting new identity” pointing towards the cells and not the arrow, since the idea is targeting the identity of new cells and not a “process”. Although in general, as a reviewer I understand the therapeutic avenues one could take in relation to plasticity, I believe the figure and related text can be made clearer to readers.

Grammar and typographical errors

There are some minor typos throughout the paper than need to be addressed by the authors, and this includes figure legends as well. Some sentences throughout the sections also need to be re written to improve clarity. For example, I had difficulty following lines 84-86, the last paragraph in section 5 (lines 154-164) and lines 254-258.

Reviewer 2 Report

This is a broad review titled ' lineage plasticity' in cancer and aims to cover the plasticity concept, its broad impact and therapeutic avenues. Overall this is an interesting review that could benefit from additional illustrations and modifications.

Major:

  • Plasticity is a broad term- thus an upfront clear definition of what terminology authors will use and what they mean in the context of the review is required. E.g., cellular identity vs, lineage plasticity, versus cellular plasticity versus histological transdifferentiation. - these are interchangeably used but technically are not the same. A diagram or schematic with these clear definitions would greatly facilitate understanding of the review.
  • Definitions of 'adaptive mechanism' are not provided- this is in fact a complex phenomenon and in the absence of a definition that the authors want readers to think about in the context of this review- it is difficult to follow and is a disservice to this complex process.- illustration here would help
  • Similarly - 'microenvironment fluctuations'-- is a terminology that mean many different things and again impacts understanding of the flow of the review.
  • In 'mechanisms' regulating lineage plasticity- authors use the word cell intrinsic and..tumors in one sentence- it is not clear if they are referring to tumor cell intrinsic, immune cell intrinsic? - 
  • Similar confusing sentence in the discussion of epigenetic and genetic alterations and plasticity. Since the discussion is about the tumor as a whole, cell type specific responses should be clearly mentioned

Minor: Some citations do not credit primary authors - for e.,g EMP was defined by developmental biologists long before the citation used. - in a similar vein- when reviews are cited more recent ones should be used as review content can be dramatically different between years.

Authors use 'VM formation' as a terminology- perhaps authors mean VM process? as VM formation refers to vascular malformations. If this is the case- the connection between vascular malformations and vascular mimicry is not clear in that section

Reviewer 3 Report

The review article, "Lineage plasticity in cancer: The tale of a skin-walker" by Thankamony et. al. provides in depth and up-to-date information about lineage plasticity among cancer. The authors have provided information about the role of Epithelial-Mesenchymal Transition (EMT) in facilitating the lineage switch. Lineage plasticity is a major challenge in tumor progression a therapeutic resistance across the type of cancers. This review certainly covers the challenges in lineage plasticity provides  insights to prepare of the development of better therapeutic strategies.

Round 2

Reviewer 2 Report

Manuscript is sufficiently improved.